

# No apparent influence of psychometrically-defined schizotypy on orientation-dependent contextual modulation of visual contrast detection

Damien J. Mannion, Chris Donkin and Thomas J. Whitford

School of Psychology, UNSW, Australia

## ABSTRACT

We investigated the relationship between psychometrically-defined schizotypy and the ability to detect a visual target pattern. Target detection is typically impaired by a surrounding pattern (context) with an orientation that is parallel to the target, relative to a surrounding pattern with an orientation that is orthogonal to the target (orientation-dependent contextual modulation). Based on reports that this effect is reduced in those with schizophrenia, we hypothesised that there would be a negative relationship between the relative score on psychometrically-defined schizotypy and the relative effect of orientation-dependent contextual modulation. We measured visual contrast detection thresholds and scores on the Oxford-Liverpool Inventory of Feelings and Experiences (O-LIFE) from a non-clinical sample ($N = 100$). Contrary to our hypothesis, we find an absence of a monotonic relationship between the relative magnitude of orientation-dependent contextual modulation of visual contrast detection and the relative score on any of the subscales of the O-LIFE. The apparent difference of this result with previous reports on those with schizophrenia suggests that orientation-dependent contextual modulation may be an informative condition in which schizophrenia and psychometrically-defined schizotypy are dissociated. However, further research is also required to clarify the strength of orientation-dependent contextual modulation in those with schizophrenia.

## INTRODUCTION

Because visual processing appears to differ in those with and without a diagnosis of schizophrenia (*Butler, Silverstein & Dakin, 2008*), increased knowledge of vision in schizophrenia can provide important progress towards our understanding of the disorder (*Yoon et al., 2013*). The concept of 'schizotypy' proposes that schizophrenia characterises the far end of a continuum of symptomatology that varies across the general population (*Lenzenweger, 2010*; *Mason & Claridge, 2015*). There is growing empirical evidence for the validity of the construct of schizotypy; for example, non-clinical individuals scoring highly on psychometric measures of schizotypy have been shown to exhibit cognitive, behavioural, and neurophysiological abnormalities similar to those observed

Corresponding author
Damien J. Mannion,
d.mannion@unsw.edu.au

in patients with established schizophrenia (*Oestreich et al., 2015*; *Asai, Sugimori & Tanno, 2008*; *Lenzenweger & O'Driscoll, 2006*; *Modinos et al., 2010*). This raises the possibility that advances in understanding schizophrenia can be made by examining how perceptual processing changes in non-clinical individuals who exhibit high levels of schizotypy (*Ettinger et al., 2015*).

However, there have been relatively few investigations into the relationship between visual perception and schizotypy. *Rawlings & Claridge (1984)* reported that those that are rated highly on schizotypy show differences in visual field asymmetries (left/right) on letter recognition and local/global tasks. *Koychev et al. (2010)* found the early visual evoked potential to be reduced in those rated highly on schizotypy. Schizotypy has also been reported to be associated with differences in visual backward masking (*Cappe et al., 2012*) and depth processing (*Barbato, Collinson & Casagrande, 2012*), although differences were limited to a particular schizotypy dimension and task, respectively. There have also been reports of an absent relationship between schizotypy and performance on perceptual organisation (*Silverstein et al., 1992*) and context effects on size and contour integration (*Uhlhaas et al., 2004*).

Visual contextual modulation, in which the perception of a target stimulus is affected by the presence of a surrounding stimulus, has received considerable investigation in those with schizophrenia (*Barch et al., 2012*; *Dakin, Carlin & Hemsley, 2005*; *Tibber et al., 2013*; *Yang et al., 2013*, for example). The orientation-dependent surround effect is a specific instance of contextual modulation that has been used to examine visual processing in those with schizophrenia (*Schallmo, Sponheim & Olman, 2015*; *Serrano-Pedraza et al., 2014*; *Seymour et al., 2013*; *Yoon et al., 2009*), and is particularly relevant due to its well-established behavioural and neural foundations. In this approach, a target pattern is surrounded by a similar pattern of either the same orientation (parallel) or the orthogonal orientation. For example, a vertical target would be surrounded by either a vertical (parallel) or horizontal (orthogonal) contextual pattern. In observers without schizophrenia, the presence of parallel context affects visual processing of the target to a much greater extent than orthogonal context (*Cannon & Fullenkamp, 1991*; *Petrov, Carandini & McKee, 2005*; *Zenger-Landolt & Heeger, 2003*). However, observers with schizophrenia are reported to be relatively less affected by the presence of a parallel compared to an orthogonal surround (*Serrano-Pedraza et al., 2014*; *Seymour et al., 2013*; *Yoon et al., 2009*; *Yoon et al., 2010*; though see *Schallmo, Sponheim & Olman, 2015*, and the Discussion).

Here, our primary aim was to examine whether such apparent alterations in visual processing in those with schizophrenia also affect those with relatively high levels of psychometrically-defined schizotypy. We adapted the paradigm of *Serrano-Pedraza et al. (2014)* to measure contrast detection thresholds for vertical patterns with parallel (vertical) and orthogonal (horizontal) surrounding context. We hypothesised that increased levels of psychometrically-defined schizotypy would be negatively related to the orientation-dependent effect of context on visual contrast detection—that is, increased scores on a psychometric measure of schizotypy would be associated with a relative decrease in the influence of context that is oriented parallel, relative to orthogonal, to a visual target. Consistent with previous reports in schizophrenia (*Serrano-Pedraza et al., 2014*,

for example), we further hypothesised that such a relationship would be attributable to a relative increase in the contrast detection threshold with orthogonal context.

Our additional aim was to advance and test a hypothesis concerning the temporal dynamics of orientation-dependent contextual modulation. Studies on auditory perception have shown that the amplitude of an early event-related potential component in response to a tone is reduced if the tone is elicited by a motor action rather than simply presented in the absence of an eliciting motor action (*Schafer & Marcus, 1973*). This reduction is weaker in those with schizophrenia (*Ford et al., 2001*) and in those that score highly on psychometrically-defined schizotypy (*Oestreich et al., 2016*). Interestingly, imposing a short delay between the motor action and the auditory tone increases the strength of the response reduction in those with schizophrenia (*Whitford et al., 2011*) and in those with high schizotypy (*Oestreich et al., 2016*). We considered that a similar phenomenon may operate in orientation-dependent contextual modulation in vision; if temporal processing is altered in those with schizophrenia, simultaneous presentation of target and context may be processed as asynchronous—a situation which strongly reduces orientation-dependent contextual modulation in typical observers (*Kilpeläinen, Donner & Laurinen, 2007*; *Petrov & McKee, 2009*). Hence, we also included a manipulation in which the onset of the surrounding context was 50 ms before the onset of the target. We predicted that this leading surround would reinstate the orientation-dependent modulation of contrast detection in those that score highly on psychometrically-defined schizotypy.

## MATERIALS & METHODS

### Participants

Participants ($N = 100$) with normal or corrected-to-normal vision were recruited from a pool of students enrolled in an introductory psychology course at UNSW Australia. Participants received course credit for their involvement and gave informed and written consent in accordance with the experiment protocols approved by the Human Research Ethics Advisory Panel in the School of Psychology, UNSW Australia (2495/153-164). All participants were naïve to the purposes of the experiment.

### Stimuli

The stimuli were similar to those used by *Serrano-Pedraza et al. (2014)* and consisted of 'context' and four potential 'target' regions. The context was a circular grating that was 20 degrees of visual angle (dva) in diameter with a spatial frequency of 1 cycle/dva, peak contrast of 25% (ramped smoothly to zero at the edges according to a raised cosine), and of horizontal or vertical orientation. The potential target regions were four circular patches that were each 3 dva in diameter and centred at 5 dva eccentricity from the middle of the context, with an opacity that ramped to zero at the edges following a raised cosine profile. On a given presentation, three of the four target regions were set to zero contrast and the remaining region displayed a vertical grating of the same spatial frequency and phase as the context and with a variable non-zero contrast (see *Design and procedure*). A small fixation marker (0.25 dva), with a centre of varying luminance and a black edge, was continually present at the centre of the display. Example stimuli are shown in Fig. 1.

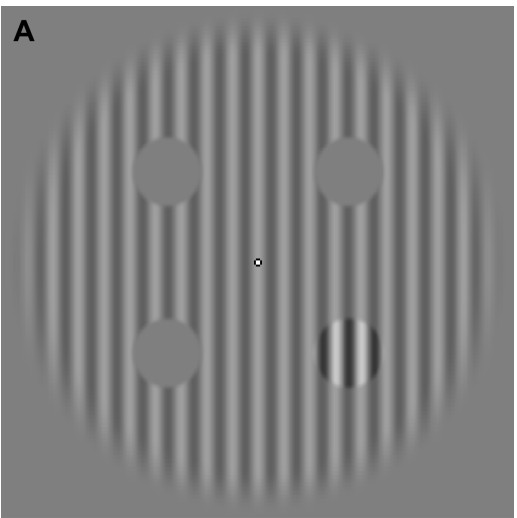

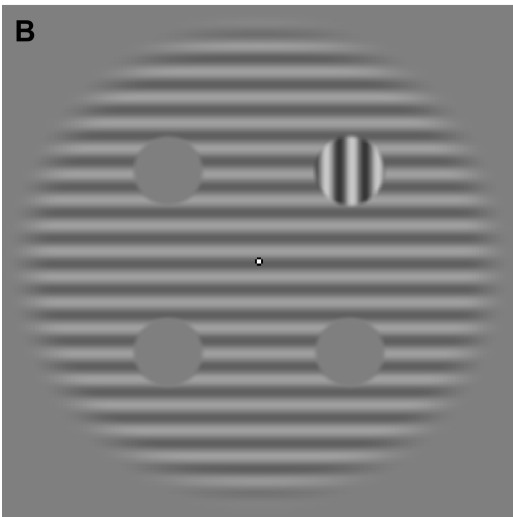

**Figure 1   Example stimuli.** (A) depicts the stimulus configuration for the parallel relative orientation condition, with the target in the bottom right quadrant. (B) depicts the stimulus configuration for the orthogonal relative orientation condition, with the target in the top right quadrant.

## Apparatus

The stimuli were presented on one of two identical Display++ LCD monitors (Cambridge Research Systems, Kent, UK) with a spatial resolution of 1,920 × 1,080 pixels, temporal resolution of 120 Hz, and mean luminance of 60 cd/m$^2$. The monitors each had a 10-bit output resolution and a linear relationship between graphics card signal and luminance. Participants viewed a monitor in one of two darkened rooms at a distance of 52 cm, via a chin rest, for a total angular subtense of 76.6 × 43.1 dva. The experiment was controlled using PsychoPy 1.82.01 (*Peirce, 2007*) and Python 2.7.10. The code and data from the study are available at https://bitbucket.org/account/user/mannionlab/projects/schizotypy_visual_contrast.

Schizotypy was assessed via the Oxford-Liverpool Inventory of Feelings and Experiences (O-LIFE; *Mason, Claridge & Jackson, 1995*). This is a paper-based questionnaire with 104 items, each requiring a yes/no response, that assess subscales of "unusual experiences," "cognitive disorganisation," "introvertive anhedonia," and "impulsive non-conformity" (see *Mason & Claridge, 2006*, for the full inventory of items). The "unusual experiences" subscale (30 items) contains items describing perceptual aberrations, magical thinking, and hallucinations, and is conceived as relating to the positive symptoms of psychosis. The "cognitive disorganisation" subscale (24 items) contains items relating to poor attention, concentration, and decision making, and is conceived as relating to the disorganisation and formal thought disorder associated with psychosis. The "introvertive anhedonia" (27 items) subscale contains items relating to anhedonia and avoidance of intimacy, and is conceived as relating to the negative symptoms of schizophrenia. The "impulsive nonconformity" (23 items) subscale contains items relating to impulsive, anti-social, and eccentric forms of behaviour. The O-LIFE has good internal consistency and test-retest reliability (*Burch, Steel & Hemsley, 1998*; *Mason, Claridge & Jackson, 1995*).

## Design and procedure

The experimental component of the study used a two-way repeated measures design, with relative orientation (parallel, orthogonal) and temporal relationship (simultaneous, leading surround) as factors. The dependent variable was the contrast detection threshold, defined as the contrast required for the spatial location of the target to be identified with 69.25% accuracy.

The experimental procedure for a given participant was conducted in a single session lasting approximately one hour. The session consisted of a series of runs, where each run measured a single condition. Each condition was measured on three different runs, giving a total of 12 runs in the session. The ordering of conditions across runs was randomised for each participant. There was a self-paced break of at least 30 s between each run.

Each run consisted of a series of trials. Each trial began with a 500 ms preparatory period in which the centre of the fixation marker was drawn in black and the remainder of the display was at mean luminance. The stimulus period was then active for 150 ms (18 frames), during which the context and target were each presented for 100 ms (12 frames). If the trial corresponded to the 'simultaneous' level of the temporal relationship factor, both the context and the target were displayed during the latter 100 ms of the stimulus period (50–150 ms). The presentation timecourse was thus that neither the context or the target was shown in the first 50 ms and then both the context and the target were simultaneously shown for 100 ms. If the trial corresponded to the 'leading surround' level of the temporal relationship factor, the context was displayed from the beginning of the stimulus period (0–100 ms) and the target was displayed during the latter 100 ms of the stimulus period (50–150 ms). The presentation timecourse was thus that the context only was shown in the first 50 ms, then both the context and the target were simultaneously shown for the next 100 ms, and then the target only was shown for the next 50 ms. The spatial phases of the context and target gratings were set to a common randomised value for each trial. On each trial, the target was randomly presented at one of the four potential spatial locations.

During presentation of the target, the centre of the fixation marker was drawn in white. Following the stimulus period, the response period was indicated by drawing the centre of the fixation marker in dark grey and by drawing thin circular outlines corresponding to the positions of the four potential target locations. The participant then indicated their selection using the '8' (top right), '7' (top left), '4' (bottom left), or '5' (bottom right) buttons on a numerical keypad. Feedback was then provided for 200 ms in the form of a 'tick' (correct) or 'cross' (incorrect) appearing in the centre of the spatial location that contained the target. If necessary, there was then a period of fixation-only presentation to enforce a minimum inter-trial interval of 2 s before commencing the next trial.

The contrast of the target on each trial was determined using a Psi adaptive staircase procedure (*Kontsevich & Tyler, 1999*). Here, contrast refers to the Michelson contrast of the grating: $(L_{max} - L_{min})/(L_{max} + L_{min})$, where $L_{max}$ and $L_{min}$ are the maximum and minimum luminances in the grating, respectively. Each staircase used a Weibull function to capture participant performance, parametrised following *Lu & Dosher (2014)* as:

$$P(c) = \zeta + (1 - \zeta - \lambda)\left(1 - e^{-(c/\tau)^{\eta}}\right). \tag{1}$$

This psychometric function describes the probability of a correct response for a given target contrast ($c$), where $\zeta$ is the chance performance level (fixed at 0.25 due to the four alternatives), $\lambda$ is the lapse rate (fixed at 0.05), $\tau$ is the threshold (the contrast corresponding to 69.25% correct performance), and $\eta$ is the slope. For the Psi procedure, the candidate contrast levels were between 0.1% and 100% in 350 logarithmically-spaced values. This distribution was also used for the threshold ($\tau$), while the slope ($\eta$) was given by 50 logarithmically-spaced levels between 0.5 and 20. There were two interleaved staircases on each run, each 40 trials in length, in randomised order across the run. Interleaved staircases were used to improve the robustness of the sampled target contrasts by reducing the opportunity for the algorithm to settle into a narrow range of contrast levels.

Before commencing the experiment, participants were introduced to the task via a set of computer-based instructions. They then completed a practice run in which the context was not present. At the conclusion of the practice run, the experimenter visually evaluated the resulting psychometric functions to determine whether participants understood the task requirements. The practice run was repeated if necessary.

Following completion of the visual task, participants completed the O-LIFE questionnaire and provided basic demographic information (complete set: gender, age, and handedness). Items on the O-LIFE questionnaire were occasionally either unanswered (1 response), answered with both response options (2 responses across 1 participant), or ambiguously answered (3 responses across 3 participants)—such items were replaced with the modal value from that subscale for that participant, after any negative scoring.

## Analysis

The experimental procedure produced 960 data points per participant, where each data point specified the contrast of the target and the correctness of the response. With four conditions, this corresponded to 240 data points per condition (3 runs per condition × 2 staircases per run × 40 trials per staircase) for each participant. We summarised the

data from each participant and condition by maximum-likelihood fitting of a Weibull psychometric function (see *Design and procedure*) to obtain threshold ($\tau$) and slope ($\eta$) values. A depiction of the raw data and the best-fitting psychometric functions are shown for a representative participant in Fig. 2, and for each participant and condition in Fig. S1.

We then evaluated the validity and reliability of the best-fitting psychometric functions for each participant. We excluded from further analysis those participants where the estimated slope was shallower than two standard deviations away from the mean slope across all participants, for any condition. This resulted in the exclusion of 7 participants, and subsequent analyses are conducted and reported based on the remaining 93 participants. The raw data and best-fitting psychometric functions for excluded participants are shown in Fig. S1.

# RESULTS

Our objective was to investigate the relationship between psychometrically-defined schizotypy and the orientation-dependent modulation of visual contrast detection. We recruited a sample of University students ($n = 93$, after exclusions) that were predominantly 18–20 years of age (81/93; see Table S1 for the complete age distribution), female (63/93), and right-handed (91/93). The distribution of O-LIFE subscale scores are qualitatively similar to the norms reported by *Mason & Claridge (2006)*, with the exception of the "unusual experiences" subscale which is lower in the current sample (see Fig. S2). The pairwise correlations between the O-LIFE subscales are shown in Fig. S3.

## Detection thresholds

The experimental paradigm was motivated by previous reports of an orientation-dependent contextual effect on contrast detection thresholds (*Petrov, Carandini & McKee, 2005* for example). First, we evaluated whether this effect was evident in the current data. As seen in Fig. 3, a simultaneously-presented parallel context resulted in a large increase in the contrast detection threshold relative to a simultaneously-presented orthogonal context; from a mean of 1.39% (95% CI [1.33, 1.45]) with orthogonal context to a mean of 25.49% (95% CI [23.23, 27.71]) with parallel context. Presenting the surrounding context slightly earlier in time than the target led to a much more moderate increase; from a mean of 1.37% (95% CI [1.31, 1.43]) with orthogonal context to a mean of 3.79% (95% CI [3.49, 4.11]) with parallel context. Hence, the current experimental paradigm was able to invoke orientation-dependent contextual modulation of contrast detection thresholds and to moderate the effect by changing the temporal schedule of the stimulation. We now consider the aims of the current study—to examine if the variabilities in the above effects are related to schizotypy.

## Relationship between contextual modulation and schizotypy for simultaneous presentation

We hypothesised that the influence of a parallel surround on the ability to perceive a target pattern, relative to an orthogonal surround, would reduce with increasing levels of psychometrically-defined schizotypy. To test this hypothesis, we compared

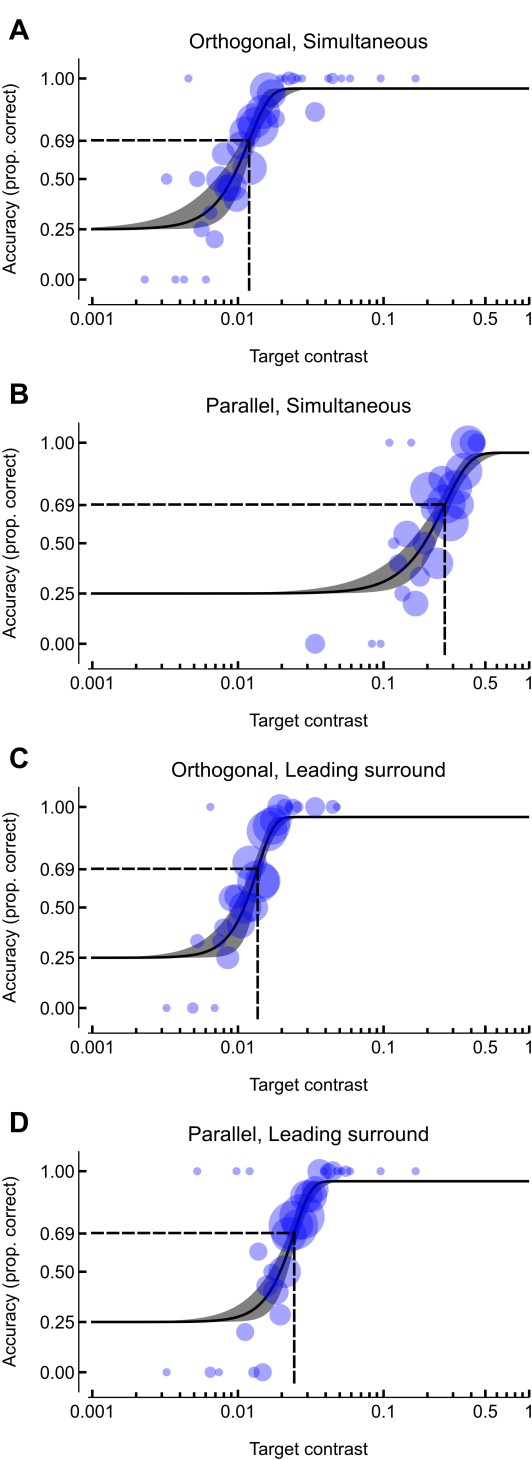

**Figure 2 Data and psychometric functions for a representative participant.** Circles represent the proportion of correct responses within a given target contrast bin, with an area that is proportional to the number of trials at that contrast. Solid black lines depict the psychometric function, with the grey surrounding region capturing the 95% CI. The dashed lines indicate the contrast detection threshold (the target contrast corresponding to 69.25% accuracy). Vertical axes are accuracy (proportion correct) and the horizontal axes are the target contrast (logarithmic spacing).

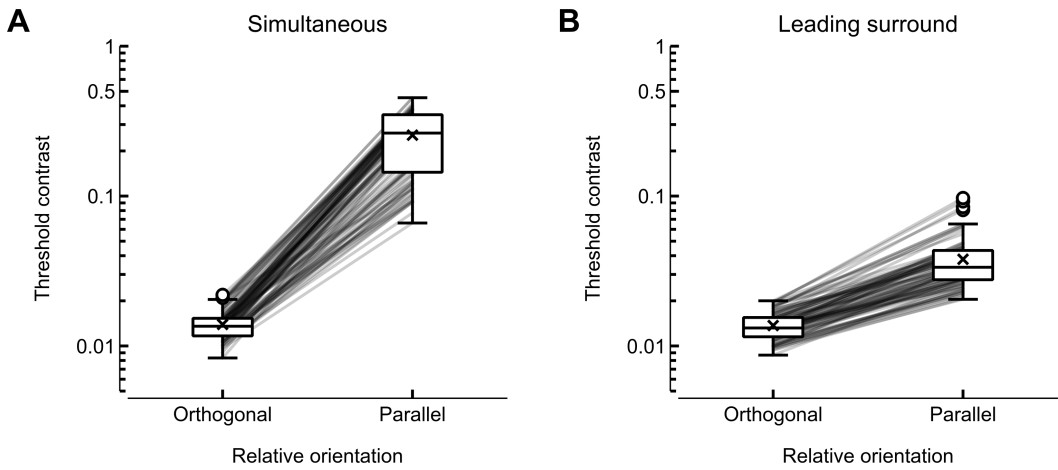

**Figure 3 Contrast detection thresholds across the four conditions.** (A) and (B) show the conditions in which the context was presented with the same (simultaneous) or different (leading surround) temporal schedule as the target. The top and bottom of a given box are the 75th and 25th percentile scores, respectively, with the dividing horizontal line at the median (50th percentile). The whiskers extend as far as the score that is within 1.5 of the inter-quartile range, and any scores outside the whiskers are marked with a circle. The crosses indicate the means of the distributions. Each line connects the thresholds for orthogonal and parallel context for a given participant.

the ranked difference in contrast detection thresholds for parallel and orthogonal surrounds with the ranked scores on each of the subscales of the O-LIFE questionnaire (see Fig. S4 for a comparison of unranked data). Contrary to our hypothesis, there was no apparent relationship between the relative magnitude of orientation-dependent contextual modulation and relative score on any of the O-LIFE subscales, as shown in Fig. 4. Quantifying the monotonicity of this relationship yielded correlation coefficients that were not significantly different from zero for any of the O-LIFE subscales ("unusual experiences": $r = 0.02$, $p = .821$, 95% CI $[-0.18, 0.23]$; "cognitive disorganisation": $r = -0.04$, $p = .681$, 95% CI $[-0.25, 0.17]$; "introvertive anhedonia": $r = -0.09$, $p = .406$, 95% CI $[-0.30, 0.13]$; "impulsive nonconformity": $r = 0.03$, $p = .766$, 95% CI $[-0.18, 0.23]$).

## Relationship between schizotypy and detection thresholds for orthogonal simultaneous context

Our next hypothesis was that the predicted reduction in orientation-dependent contextual modulation of contrast detection thresholds in those that scored highly on psychometrically-defined schizotypy would be due to elevated contrast detection thresholds with an orthogonal context. This would be expressed as a positive relationship between the ranked scores on psychometrically-defined schizotypy and the ranked contrast detection thresholds in the orthogonal context condition with simultaneous presentation. However, we found little evidence for a monotonic relationship between these measures on any of the O-LIFE subscales ("unusual experiences": $r = 0.07$, $p = .521$, 95% CI $[-0.15, 0.27]$; "cognitive disorganisation": $r = -0.05$, $p = .646$, 95% CI $[-0.26, 0.16]$; "introvertive
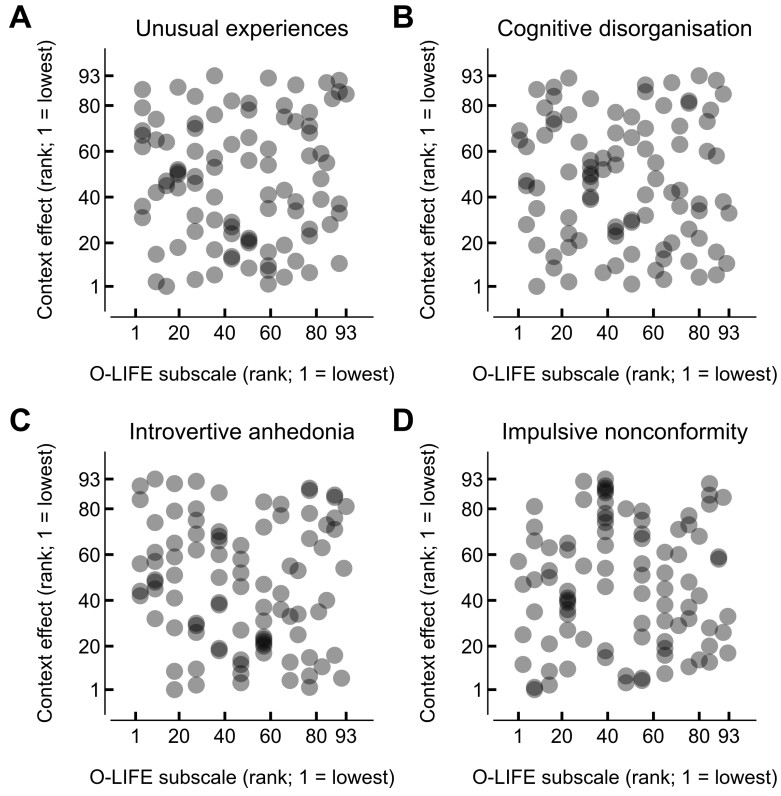

**Figure 4  Comparison of ranked O-LIFE score and the ranked magnitude of the orientation-dependent effect of context during simultaneous presentation.** Each point shows a single participant's O-LIFE subscale rank and orientation-dependent context effect (obtained by ranking the difference between the contrast detection threshold with a parallel surround and that with an orthogonal surround, both during simultaneous presentation).

anhedonia": $r = -0.11$, $p = .314$, 95% CI $[-0.30, 0.10]$; "impulsive nonconformity": $r = 0.04$, $p = .724$, 95% CI $[-0.18, 0.24]$). These relationships are shown in Fig. S5.

## Relationship between schizotypy and temporal influences on contextual modulation

An additional aim of this experiment was to investigate the temporal characteristics of orientation-dependent contextual modulation and its relationship with psychometrically-defined schizotypy. In particular, we hypothesised that presenting the context slightly earlier in time than the target would yield an increased influence of a parallel context, relative to an orthogonal context, for those that scored highly on psychometrically-defined schizotypy. Accordingly, we examined the relationship between the ranked level of psychometrically-defined schizotypy and the relative magnitude of the difference between contrast detection thresholds with parallel and orthogonal context for the 'leading surround' condition. However, there was no discernible relationship between these measures on any of the O-LIFE subscales ("unusual experiences": $r = 0.14$, $p = .177$, 95% CI $[-0.08, 0.36]$; "cognitive disorganisation": $r = -0.01$, $p = .933$, 95% CI $[-0.22, 0.21]$; "introvertive

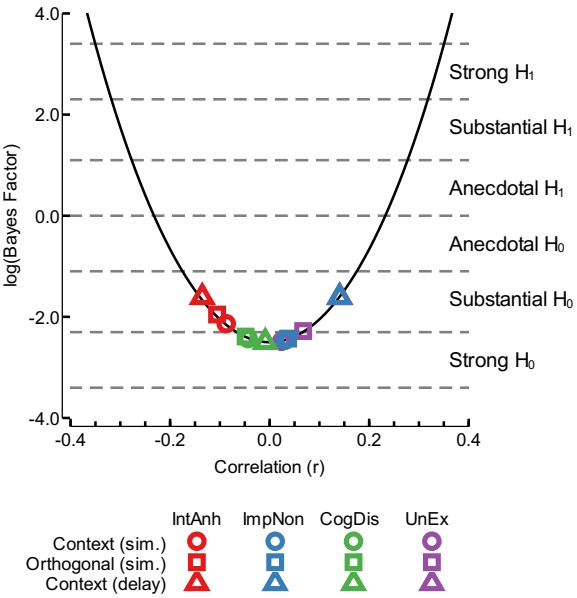

**Figure 5  Bayesian analysis of correlation coefficients.** We used the approach of *Wetzels & Wagenmakers (2012)* to determine the relationship between a given correlation coefficient and its Bayes factor, with a fixed sample size of 93, as depicted by the solid black line. The points show the positioning on this curve of the analyses conducted in this study; different colours depict the dimensions of the schizotypy scale (red: introvertive anhedonia (IntAnh); blue: impulsive nonconformity (ImpNon); green: cognitive disorganisation (CogDis); purple: unusual experiences (UnEx)) and different shapes depict the class of analysis (circles: context effect with simultaneous presentation; squares: orthogonal threshold with simultaneous presentation; triangles: context effect with delayed target presentation). Dashed grey lines denote the Bayes factor interpretation categories, following *Wetzels & Wagenmakers (2012)*.

anhedonia": $r = -0.14$, $p = .196$, 95% CI $[-0.33, 0.07]$; "impulsive nonconformity": $r = 0.14$, $p = .179$, 95% CI $[-0.07, 0.34]$), as shown in Fig. S6.

## Bayesian analyses

The above analyses show that there is a high probability of observing the obtained correlation coefficients if the null hypothesis of no monotonic relationship is true. However, this failure to reject the null hypothesis does not allow for strong conclusions concerning the viability of the null hypothesis itself. To evaluate the strength of the evidence in favour of the null hypothesis, we used the approach of *Wetzels & Wagenmakers (2012)* to compute Bayes factors for each of the obtained correlations given our sample size ($n = 93$). As shown in Fig. 5, the obtained correlation coefficients have Bayes factors that are consistent with the null hypothesis (no monotonic relationship between the given variables); the logarithm of all of the Bayes factors was $<0$ (highest log Bayes factor $= -1.61$). Each of these Bayes factors satisfies the criteria for "substantial" or "strong" support for the null hypothesis, according to the definitions of *Wetzels & Wagenmakers (2012)*.

## DISCUSSION

The primary aim of this study was to determine whether psychometrically-defined schizotypy relates to the extent to which the ability to detect a visual pattern depends

on the relative orientation of the target and its surrounding context. Based on previous reports that a diagnosis of schizophrenia is associated with a reduction of the impact of parallel context, relative to orthogonal context (*Serrano-Pedraza et al., 2014*; *Yoon et al., 2009*), we hypothesised that individuals scoring highly on psychometrically-defined schizotypy dimensions would be less affected in their ability to detect a vertical pattern with a vertical surround relative to a horizontal surround. However, contrary to our hypothesis, we found no evidence of such a relationship—instead, we found evidence for the absence of such a relationship.

We also investigated whether the threshold for detecting a visual pattern with an orthogonal surround was related to the relative score on psychometrically-defined schizotypy. Previous reports have suggested that those with schizophrenia have elevated thresholds under such presentation conditions (*Serrano-Pedraza et al., 2014*; *Yoon et al., 2009*). However, we again find no evidence of such a relationship—furthermore, we again found evidence for the absence of such a relationship.

Our secondary aim in conducting this study was to determine whether the effect of the temporal presentation schedule of the target and surround affected visual contrast detection thresholds differently depending on the relative degree of psychometrically-defined schizotypy. Specifically, we predicted that those ranked highly on psychometrically-defined schizotypy would be more affected by the slightly-earlier presentation of the surround relative to the target. However, contrary to this hypothesis, we find no evidence of a relationship between psychometrically-defined schizotypy and orientation-dependent contextual modulation when the surround led the presentation of the target—we instead found evidence for the absence of such a relationship.

## Implications of the current findings

The results of the current study are consistent with psychometrically-defined schizotypy being unrelated to the orientation-dependent modulation of visual contrast detection. Given that previous studies report that the extent of such orientation-dependent contextual modulation is related to a diagnosis of schizophrenia (*Serrano-Pedraza et al., 2014*; *Yoon et al., 2009*), this suggests that orientation-dependent modulation of visual contrast detection may be a situation in which schizophrenia and psychometrically-defined schizotypy are dissociated. As stated by *Ettinger et al. (2015)*, such situations can be indicative of "protective or compensatory mechanisms" (p. S418) and are hence of considerable interest in understanding the transition to psychosis. Orientation-dependent contextual modulation may be a particularly useful paradigm for revealing such mechanisms due to its well-studied behavioural and neural foundations (*Seymour et al., 2013*; *Yoon et al., 2013*) and established theories regarding its circuitry in schizophrenia (*Yoon et al., 2010*). Should the results and conclusions presented here prove to be robust, future studies using schizotypy may usefully correspond and interact with investigations of schizophrenia to aid in clarifying the mechanisms that selectively relate to the transition to disorder.

## Potential limitations of the current study

However, we must also consider whether there are factors relating to the current study that hamper our ability to draw strong conclusions. As we require the presence of the

orientation-dependent contextual modulation of visual contrast detection in order to probe its potential relationship with psychometrically-defined schizotypy, it is important to consider whether this requirement was satisfied in the current experiment. As shown in Fig. 3, the presence of parallel context led to a large increase in contrast detection thresholds compared to an orthogonal context—the magnitude of this increase is comparable to that reported by *Serrano-Pedraza et al. (2014)* using a similar paradigm.

A further important prerequisite is that the sample contained sufficient variation in levels of psychometrically-defined schizotypy. The sampled O-LIFE scores were qualitatively similar to the general-population norms reported by *Mason & Claridge (2006)*, with the exception of the "unusual experiences" subscale in which the current sample was lower than the norms. It is possible that the current sample was insufficiently variable along this dimension to capture a true relationship with orientation-dependent contextual modulation. Furthermore, it is also possible that the current sample contained insufficient participants with extreme scores. Future studies that use a prescreening recruitment strategy may be useful in clarifying the relationship between schizotypy and orientation-dependent contextual modulation.

A desirable component of an investigation of schizotypy, absent in the current study, is the inclusion of a positive control condition—that is, a condition in which there is a strong expectation, based on an accumulation of previous findings, that performance on that condition will be related to psychometrically-defined schizotypy. Recovering such a result in the observed data would provide increased confidence in the validity of the analyses and on the sufficiency of the sample characteristics. However, we suggest that such an approach is, unfortunately, premature for vision and schizotypy—as reviewed in the Introduction, there are currently few (if any) established paradigms that are known to reliably associate visual performance with psychometrically-defined schizotypy. The identification and development of such paradigms is an important area of future research.

Finally, an additional potential limitation of this study (and of related studies) concerns the parameterisation of the surrounding stimulus contrast. Here, the contrast of the surround was fixed (at 25%) for all participants—similar to the usage of fixed surround contrasts of 25% and 100% in *Serrano-Pedraza et al. (2014)* and *Yoon et al. (2009)*, respectively. When comparing groups with differing contrast sensitivity, such as is evident in those with and without schizophrenia (*Serrano-Pedraza et al., 2014*), the fixed nature of the surround contrast becomes potentially challenging. This challenge arises because the surround pattern will have different *effective* contrasts for the groups—for those that are highly sensitive to visual contrast, a surround of a particular stimulus contrast will have a greater effective contrast than it would for those that have reduced contrast sensitivity. Particularly given that the relationship between surround contrast and its effect on the perception of the central target is nonlinear (*Petrov, Carandini & McKee, 2005*), it is difficult to identify whether it is changes to the nature of the centre-surround relationship that explain the differing visual performance between groups or whether it instead reflects that the groups are being evaluated at differing locations on a similar centre-surround relationship. These candidate explanations have differing implications for the underlying mechanisms, and we suggest that future studies be devoted to evaluating and resolving this

uncertainty—as has been investigated for contextual modulation changes with aging by *Karas & McKendrick (2011)*.

## Evaluating the evidence for the effect in schizophrenia

It is also important to assess the strength of the previously-reported evidence for the differential orientation-specific contextual modulation in schizophrenia before we can confidently assert the distinction between schizophrenia and schizotypy that is suggested by our results. The hypothesis that we entertained in the current study was motivated by previous reports of a different profile of orientation-dependent contextual modulation of contrast perception in schizophrenia—in particular, the studies by *Yoon et al. (2009)* and *Serrano-Pedraza et al. (2014)*.

The fundamental prediction of the proposed alteration of orientation-dependent contextual modulation of contrast perception in schizophrenia is that there will be an interaction between group (those with and without a diagnosis of schizophrenia) and stimulus condition (parallel and orthogonal context orientation) for the relevant dependent variable. In *Yoon et al. (2009)*, the primary dependent variable was the contrast by which a section of a target annulus had to be reduced for participants to perform with 79% accuracy (percent correct) on a yes/no task in which they judged whether there was a section of reduced contrast in the target. While the relevant statistical test for the interaction between group and stimulus condition (parallel and orthogonal surround) was not reported in *Yoon et al. (2009)*, a re-analysis (the raw data was made available by the authors at http://doi.org/10.5281/zenodo.163785) of the data indicated that this interaction is statistically significant ($F_{1,35} = 5.85$, $p = 0.021$). In *Serrano-Pedraza et al. (2014)*, the primary dependent variable was the contrast at which the spatial location of a target could be identified from four alternatives at 62% accuracy (percent correct). The interaction between group and stimulus condition was not statistically significant, providing no evidence for altered orientation-dependent contextual modulation of contrast perception in schizophrenia. Finally, there is also a relevant study by *Schallmo, Sponheim & Olman (2015)* in which the primary dependent variable was the contrast by which an isolated target needed to be adjusted in order to perceptually match the contrast of an adjacent target that was embedded in an articulated spatial context. The interaction between group (those with and without a diagnosis of schizophrenia) and stimulus condition (parallel and orthogonal surround) was not statistically significant.

The evidence considered thus far for a difference in orientation-dependent contextual modulation of target perception in those with and without a diagnosis of schizophrenia is equivocal—while the data reported in *Yoon et al. (2009)* show a significant interaction between group (those with and without a diagnosis of schizophrenia) and surround orientation (parallel and orthogonal), the data reported in *Serrano-Pedraza et al. (2014)* and *Schallmo, Sponheim & Olman (2015)* do not. However, *Yoon et al. (2009)* and *Serrano-Pedraza et al. (2014)* also consider a transformed dependent variable in which the performance with parallel and orthogonal context is expressed relative to performance where the target has no articulated surrounding context. In both cases, statistical analysis on such ratios (or log ratios, in *Serrano-Pedraza et al. (2014)*) demonstrated a significant

interaction between group and stimulus condition—with the parallel ratio significantly larger in those without a diagnosis of schizophrenia and the orthogonal ratio not significantly different in those with and without schizophrenia.

The nature of the dependent variable thus appears to be relevant to the conclusions drawn about the presence of altered orientation-dependent contextual modulation of contrast perception in schizophrenia. The rationale behind the ratio transformation appears to be based on a desire to adjust for differences in performance in the absence of any articulated context. However, interpreting ratios can be challenging due to a critical requirement to be satisfied in order for the transformation to appropriately achieve the desired control without distortion—specifically, the relationship between the numerator and the denominator must be linear and must pass through the origin (*Allison et al., 1995*; *Curran-Everett, 2013*). We re-analysed the data of *Yoon et al. (2009)* and *Serrano-Pedraza et al. (2014)* to assess these requirements (see Article S1). We find that the requirements were not always satisfied, and that such violations had demonstrable consequences in the *Serrano-Pedraza et al. (2014)* data while the *Yoon et al. (2009)* data appeared to be less affected. Hence, we suggest that the comparison between those with and without schizophrenia using such transformed data needs to be interpreted with caution (see *Curran-Everett, 2013* for further discussion of such issues).

## CONCLUSION

Psychometrically-defined schizotypy appears to be unrelated to the degree to which visual contrast detection thresholds are affected by the relative orientation between the target and its surround. This could identify orientation-specific contextual modulation of visual contrast detection as a paradigm in which schizophrenia is dissociated from psychometrically-defined schizotypy, and hence important in identifying protective or compensatory mechanisms. However, additional research is required to affirm the reported relationship between orientation-dependent contextual modulation and schizophrenia.

## ACKNOWLEDGEMENTS

Thanks to Norah Grewal for data collection. We also thank the authors of *Yoon et al. (2009)*, *Serrano-Pedraza et al. (2014)*, and *Schallmo, Sponheim & Olman (2015)* for sharing the data from their studies.

### Funding
This research was supported in part by grants from the National Health and Medical Research Council of Australia (APP1090507 to TW) and the Australian Research Council (DP140104394 to TW and DE130100129 to CD). The funders had no role in study design, data collection and analysis, decision to publish, or preparation of the manuscript.

### Grant Disclosures
The following grant information was disclosed by the authors:
National Health and Medical Research Council of Australia: APP1090507.
Australian Research Council: DP140104394, DE130100129.

### Competing Interests
The authors declare there are no competing interests.

### Author Contributions
- Damien J. Mannion conceived and designed the experiments, analyzed the data, wrote the paper, prepared figures and/or tables, reviewed drafts of the paper.
- Chris Donkin contributed reagents/materials/analysis tools, reviewed drafts of the paper.
- Thomas J. Whitford conceived and designed the experiments, contributed reagents/materials/analysis tools, reviewed drafts of the paper.

### Human Ethics
The following information was supplied relating to ethical approvals (i.e., approving body and any reference numbers):
1. Human Research Ethics Advisory Panel in the School of Psychology, UNSW Australia.
2. Approval #2495/153-164.

### Data Availability
The code and data are available in repositories at https://bitbucket.org/account/user/mannionlab/projects/schizotypy_visual_contrast.

### Supplemental Information
Supplemental information for this article can be found online at http://dx.doi.org/10.7717/peerj.2921#supplemental-information.

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
