# Peer review of "No apparent influence of psychometrically-defined schizotypy on orientation-dependent contextual modulation of visual contrast detection"

_PeerJ, doi:10.7717/peerj.2921_

## Round 0.1 · original submission · Major Revisions

The reviewers had major concrns about the study and interpretation of data, but felt the experiments were well designed. Per PeerJ policy, articles are potentially publishable so long as they are experimentally sound. Therefore, if you address the concerns of the reviewers, we would be happy to re-review the study.

·

Basic reporting

The manuscript is well written and the figures are informative and unambiguous. No further comments.

Experimental design

The experiments are well documented and the experimental design is well constructed to answer the experimental question posed in this study. No further comments.

Validity of the findings

The manuscript is well written and the experiments are well documented. Unfortunately, the foundation upon which this study is built is flawed; it is based upon studies of orientation-dependent contextual modulations conducted by other groups (Yoon et al. (2009) and Serrano-Pedraza et al. (2014)), who observed deviations in visual perception in a sample of schizophrenic patients, with respect to neurotypical individuals.
The central research question, posed in this current study, whether this non-typical perception is also observed in non-clinical individuals with a high degree of schizotypy, might be meaningless because of grave concerns regarding the reliability of the results reported in the aforementioned studies: Yoon et al. (2009) and Serrano-Pedraza et al. (2014). Those concerns were documented by the authors themselves in the subsection “Evaluating the evidence for the effect in schizophrenia”

The negative outcome reported in this manuscript implies that orientation-dependent contextual modulation of contrast perception is altered in patients with schizophrenia, but not in non-clinical individuals with a high score in schizotypy. The relevance of this potentially interesting finding is however completely overshadowed by the doubts regarding the validity of the studies upon which this research is based. The authors, likely surprised by the unexpected outcome of their own study, did take a second look at those studies and provided compelling arguments that casts doubt over the conclusions reported in those earlier studies. The authors were correct to express those doubts in their manuscript; but unfortunately, because the evidence for the effect in schizophrenia is questionable, it also undermines the relevance of their own research.

Because the authors speculated that there would be a significant negative correlation between an individual’s score on a psychometric measure of schizotypy and his/her ability to detect the target during the visual contextual modulation task paradigm they adapted from of Serrano-Pedraza et al. (2014), they also introduced some extra experimental conditions; those conditions differed only slightly from the original ones because of a small manipulation, engineered to affect temporal processing and aimed to abolish the expected relationship between the ability to detect the target and the degree of schizotypy. Since the relationship in question could never be established in the first place, the results yielded by those extra two conditions are unfortunately totally irrelevant.

The relationship between orientation-dependent contextual modulation of visual contrast detection and its potential relationship with psychometrically defined schizotypy might of course have been obscured as the authors pointed out because “the sample contained insufficient participants with extreme scores.” A quick assessment that might verify whether this concern is justified or not, can be obtained by repeating the analysis including only the 1/3 of individuals with the highest schizotypy scores and the 1/3 with the lowest scores. Based upon visual inspections of the graphs in figure 4, it seems however very unlikely that the outcome of this extra analysis would tip the balance even slightly in favor of a different outcome.

Taken together, the authors report an outcome that given the unreliability of the studies it is based upon is not very informative. In addition, they engineered two extra conditions that never could be put to a real test because it was meant to abolish the effect of an expected outcome that was never observed.

Additional comments

The question whether the effect addressed in this study, is reliable in schizophrenia needs to be addressed first. Once this question is resolved, then the results presented in this current study can be used independently of the outcome: either as extra evidence that the effect does not exist (no effect) or to report differences in visual perception between patients diagnosed with schizophrenia and individuals with high scoring schizotypy (if there is an effect).

·

Basic reporting

The article is well-written and clear. Contains all relevant references to previous literature on the topic, as far as I can tell.

Experimental design

The experimental design is appropriate to answer the question. I think that it would be good to assess/report the test-retest reliability of the measures that are correlated, as this limits the power to discover correlations between them.

Validity of the findings

I wonder about the use of sample ranks in describing the correlations in figure 4 and in the statistical significance tests described e.g., on line 199-200. Why are the raw scores not used here? Wouldn't the assumptions of the statistical test for significance of the Pearson correlation coefficient be violated by inducing a near-uniform distribution on the scores? How does this choice affect the analysis in Figure 5? I don’t necessarily think that you would find a correlation with the raw scores, but I would have liked to see these scores anyway, because their distribution might shed light on the interpretation of the results.

The discussion concludes with a section that questions whether previous findings in the literature of differences in orientation-selective surround suppression between patients with schizophrenia . In particular, the authors question whether there was an interaction between condition and group in our previous work (Yoon et al. 2009). To allow the authors to assess this interaction, I have made the data publicly available here: https://github.com/arokem/yoon2009-schizophrenia-bulletin, and provide an analysis that demonstrates this interaction here: https://github.com/arokem/yoon2009-schizophrenia-bulletin/blob/master/Yoon2009.ipynb

I have also provided the authors with the analysis required in order to assess whether the use of ratios was appropriate in our original paper. I am not entirely sure what to look for here, but it seems that some kind of formal test is required here, as well as in their assessment of the data from Serrano-Pedraza et al.'s work in supplemental Figure 6 (the authors state that "If the assumptions for a valid ratio transform were met, no relationship would be evident in the centre and right columns”, but no formal test of this is presented). Furthermore, the presentation of Schallmo et al.’s results in this section makes it seem as though they found no difference between controls and patients, where such a difference was found, using a normalized contextual modulation index. Was use of this normalized index justified in that case?

Another interpretation of these results is that the personality measure of schizotypy does not capture the biological variability that is captured by orientation-selective surround suppression, and that OSSS taps some other critical biological factors that distinguish high-schizotypy individuals from those that develop schizophrenia symptoms (this is also mentioned in the Ettinger review cited). I think that it would be appropriate to present this alternative interpretation as well.

Additional comments

The link to Bitbucket on L88 is broken because of the line-break, and reflow with L89.

L 156: How occasionally were items unanswered?

L325: typo: “an” => “a"

---

## Round 0.2 · accepted · Accept

Please note that there remain a few minor reviewer comments. Please address these fully in the final manuscript during production.

·

Basic reporting

no comment

Experimental design

no comment

Validity of the findings

Comment:

Because the evidence that "the effect" in those with schizophrenia is questionable at best, the parsimonious explanation for the results obtained by the authors, is that there is no differential orientation specific contextual modulation in schizophrenia nor in psychometrically-defined schizotypy.
In the abstract, however, the authors only mention the tantalizing possibility that orientation-dependent contextual modulation may be “an informative condition in which schizophrenia and psychometrically-defined schizotypy are dissociated”.
For completeness, the authors should mention both possible implications of their findings in the abstract.

·

Basic reporting

no comment

Experimental design

no comment

Validity of the findings

The additional analysis addressing the validity of the use of ratios in previous studies is appreciated (and a useful example for future practitioners). The authors have addressed all my comments here.

Additional comments

Regarding Schallmo et al.'s results, I owe the authors an apology: I confused Schallmo et al.’s 2013 study (in PLoS One), with the 2015 study mentioned by the authors. The 2013 study may also be considered relevant to these questions (and found a difference between patients and controls). I think it's worth mentioning (at least in the introduction), but the authors may choose to leave it out, because it possibly pertains to other perceptual phenomena (contour completion, rather than contrast perception). Either way, I don’t have any objections to the way that the 2015 study is described, and I apologize for my confusion.

Regarding the results from our previous studies, I think that because we have made the data publicly available, it would be appropriate for the authors to link to the data on Github/Zenodo, so that future readers of their article can also easily find these data.